# Improved measurement of disease progression in people living with early Parkinson's disease using digital health technologies

Matthew D. Czech [1]✉, Darryl Badley[1], Liuqing Yang[1], Jie Shen[1], Michelle Crouthamel [1], Tairmae Kangarloo[2], E. Ray Dorsey [3], Jamie L. Adams[3] & Josh D. Cosman [1]

## Abstract

**Background** Digital health technologies show promise for improving the measurement of Parkinson's disease in clinical research and trials. However, it is not clear whether digital measures demonstrate enhanced sensitivity to disease progression compared to traditional measurement approaches.
**Methods** To this end, we develop a wearable sensor-based digital algorithm for deriving features of upper and lower-body bradykinesia and evaluate the sensitivity of digital measures to 1-year longitudinal progression using data from the WATCH-PD study, a multicenter, observational digital assessment study in participants with early, untreated Parkinson's disease. In total, 82 early, untreated Parkinson's disease participants and 50 age-matched controls were recruited and took part in a variety of motor tasks over the course of a 12-month period while wearing body-worn inertial sensors. We establish clinical validity of sensor-based digital measures by investigating convergent validity with appropriate clinical constructs, known groups validity by distinguishing patients from healthy volunteers, and test-retest reliability by comparing measurements between visits.
**Results** We demonstrate clinical validity of the digital measures, and importantly, superior sensitivity of digital measures for distinguishing 1-year longitudinal change in early-stage PD relative to corresponding clinical constructs.
**Conclusions** Our results demonstrate the potential of digital health technologies to enhance sensitivity to disease progression relative to existing measurement standards and may constitute the basis for use as drug development tools in clinical research.

## Plain language summary

Parkinson's disease can impact a person's ability to move, which can result in slow or rigid movements. Wearable sensors can be used to measure these symptoms and could be particularly useful to detect changes early in the course of the disease when symptoms may be subtle. We developed a wearable sensor-based method to measure movement in people with early Parkinson's disease that uses wrist and foot-worn sensors. Our results demonstrate that our sensor-based measurements can accurately quantify progressive changes in movement function. Such measurements may allow researchers to more accurately evaluate how well treatments designed to slow the course of Parkinson's disease are working in the future.

Given the slow evolution of progressive neurological disorders there is a need to improve the longitudinal measurement of signs and symptoms to allow for more effective patient selection and monitoring in clinical trials of novel therapies. This is especially true in Parkinson's disease (PD), where few objective biomarkers for diagnosis, progression, and disease severity currently exist[1]. There is increasing focus on the development of disease-modifying treatments designed to slow or stop progression of disability, and intervening at the earliest stages of disease may enable prevention of long-term, irreversible neural damage[2]. However, measuring changes in subtle signs and symptoms early in the course of PD is challenging with existing instruments such as the Movement Disorder Society Unified Parkinson's Disease Rating Scale (MDS-UPDRS) due to inter- and intrarater variability as well as low temporal resolution[3–5].

PD serves as a potent test-case for evaluating the promise and utility of digital health technologies (DHTs) for longitudinal monitoring given that early cardinal motor signs may be detected using inertial measurement unit

---

[1]AbbVie, North Chicago, IL, USA. [2]Takeda Pharmaceutical Company Limited, Cambridge, MA, USA. [3]University of Rochester Medical Center, Rochester, NY, USA. ✉e-mail: Matt.Czech@abbvie.com

(IMU)-based wearable sensors[6]. DHTs provide a unique opportunity to derive objective, high resolution, and less variable endpoints for therapeutic trials in early PD, painting a fuller and more meaningful picture of disease burden[7]. Ultimately, this may lead to more sensitive measures of functional impairment and response to treatment enhancing the efficiency of longitudinal clinical research and trials.

Research using wearable sensors to measure motor symptoms in PD has expanded considerably in recent years, especially as commercial devices including IMUs have become more widely available[8]. Various studies have demonstrated validity of DHTs in measuring the signs and symptoms of PD compared to ground truth systems and clinical constructs, as well as measuring response to symptomatic therapies[9–11]. For example, Lipsmeier et al.[12] demonstrated adherence, reliability, and construct validity of a smartphone- and smartwatch-based DHT that measures symptoms across a variety of domains, including bradykinesia, bradyphrenia and speech, tremor, gait and balance in early PD[12]. Similarly, Burq et al.[13] demonstrated adherence, reliability, convergent validity, and sensitivity to treatment for a smartwatch-based virtual exam to measure severity of upper-extremity bradykinesia, rest tremor, and gait[13]. However, there has been limited focus on the sensitivity of DHT-based measures to changes in function over time[14,15], a critical component of adequate clinical validation especially in progressive disorders[16–19]. Moreover, few studies have examined comparisons between digital measures and standard clinical measures in their ability to detect changes in PD motor signs in the earliest stages of disease.

In the current work, we focus on one cardinal motor symptom of PD, bradykinesia, to investigate the sensitivity of digital measurement relative to traditional clinical assessment in measuring longitudinal changes in early-stage PD patients. For bradykinesia symptomatology, several wearable-based methods have been proposed and validated for at-home[20–27] and assessment-based[28–31] measurement. However, few are openly available for researchers to validate or improve using external datasets, and none have been used to investigate sensitivity to longitudinal disease progression[32,33]. To this end, we describe and make available heuristic algorithms for extracting clinical features of upper and lower extremity bradykinesia during pronation-supination and toe-tapping tasks using an IMU-based device worn on the wrist or foot, respectively. We demonstrate clinical validity of this approach by verifying relationships with appropriate clinical constructs. Importantly, we demonstrate enhanced sensitivity of digital measures relative to corresponding MDS-UPDRS items and sub-scores to distinguish change over a 1-year period in early-stage PD patients. Our results provide preliminary evidence to support enhanced sensitivity of digital measurements in their ability to detect changes in PD motor signs over time compared to traditional clinic-based assessment.

## Methods
### Participants and procedure
The multicenter, observational WATCH-PD (Wearable Assessment in The Clinic and at Home in PD) (NCT03681015) study recruited 82 early, untreated PD patients and 50 age-matched non-PD participants over the age of 30 to take part in a variety of gait, bradykinesia, and tremor related tasks both in-clinic and at home over the course of a 12-month period (see Adams et al.[11]). Participants were recruited from clinics, study interest registries, and social media and enrolled at 17 Parkinson Study Group research sites. A limitation of the study population is that participants were largely white and well-educated, however the demographic profile of participants mirrored that of larger, similar observational studies in PD on which the current study was modeled[34]. Principal inclusion criteria for participants with PD were age 30 or greater at diagnosis, disease duration less than 2 years, and Hoehn and Yahr stage two or less. Exclusion criteria included baseline use of dopaminergic or other PD medications and an alternative parkinsonian diagnosis. The study was powered to detect a mean change in MDS-UPDRS Part III of 6.9 over 12 months and yield 30 participants completing the study off medication, accounting for up to half of participants to begin dopaminergic therapy and 15% drop out.

In-clinic assessments were performed at baseline and 1, 3, 6, 9, and 12 months. Participants were instrumented with 5 APDM Opal wearable sensor devices (Clario, Inc.), placed on both feet and wrists, and on the lower back[35]. Investigators completed and scored all components of the MDS-UPDRS while devices contemporaneously collected triaxial accelerometer, gyroscope, and magnetometer data with a sampling rate of 128 Hz. The analysis presented was limited to data from pronation-supination and toe-tapping bradykinesia assessments from part III of the MDS-UPDRS, given that these were the only tests that could be reasonably expected to generate a robust signal with foot and wrist mounted sensors. Total bradykinesia score was calculated by summing MDS-UPDRS items 3.4–3.9 and 3.13–3.14.

Following removal of participant assessments with missing clinical scores, technical problems with data capture, or less than a threshold of 5 movements detected, data from 76 PD participants (age = 64.1 ± 9.4 years, 32 F, body mass index (BMI) = 27.1 ± 4.7) and 40 non-PD participants (age = 60.8 ± 10.1 years, 23 F, BMI = 27.0 ± 8.0) were included in the pronation-supination analysis at baseline. At the final 12-month visit, due to the reasons mentioned above as well as participant drop out, data from 52 PD and 35 non-PD participants were available for analysis. For the toe-tapping analysis, data from 78 PD patients and 39 non-PD participants were available at baseline for analysis. At the final 12-month visit, data from 56 PD patients and 35 non-PD participants were available for analysis.

### Ethics
The WCG™ Institutional Review Board approved (IRB Tracking #: 20183288) the procedures used in the study, and there was full compliance with human experimentation guidelines. All participants provided written informed consent before study participation.

### Feature extraction
Device orientation data (Euler angles) from sensors placed on the wrist and foot on the most affected side during pronation-supination and toe-tapping tasks, respectively, were used to derive digital features from PD patients. In non-PD participants, sensor data from the dominant hand was used for the analysis, as determined by the Edinburgh Handedness Inventory.

In order to mirror the key attributes scored by clinicians during the MDS-UPDRS Part 3 clinician ratings, signal processing-based feature extraction focused on the domains of speed, amplitude, rhythm, slowing of movement, and decrementing amplitude (Fig. 1). Specifically, amplitude, frequency, and max velocity were derived for each pronation-supination and toe-tapping movement, and summarized across each task using median, variability, and slope statistics (Fig. 1). Each movement was detected by band pass filtering (0.3–20 Hz second-order Butterworth filter) raw Euler angle data, spline interpolating, and implementing a peak detection algorithm, similar to the method previously described by Martinez-Manzanera et al.[29]. Pseudocode is available in the Supplementary Data.

### Statistics and reproducibility
Composite digital scores for pronation-supination and toe-tapping tasks were derived using unweighted z-score summation. Specifically, three summary metrics (median, standard deviation, and slope) were used to summarize each of three individual movement features (frequency, amplitude, and max velocity) across each assessment. Normalized z-scores are calculated for each of the resulting 9 features per assessment (Supplementary Table 1). Z-scores are estimated by normalizing each feature by the corresponding feature in non-PD controls from the baseline visit (Eq. (1)). For example, in the case of slope frequency, slope frequency for each assessment is subtracted by the mean of the slope frequency in non-PD controls at baseline and the resulting value is divided by the standard deviation of slope frequency in non-PD controls at baseline:

$$Z \text{ score} = (\text{feature} - \text{mean(feature in nonPD at baseline)}) / \text{sd(feature in nonPD at baseline)} \tag{1}$$

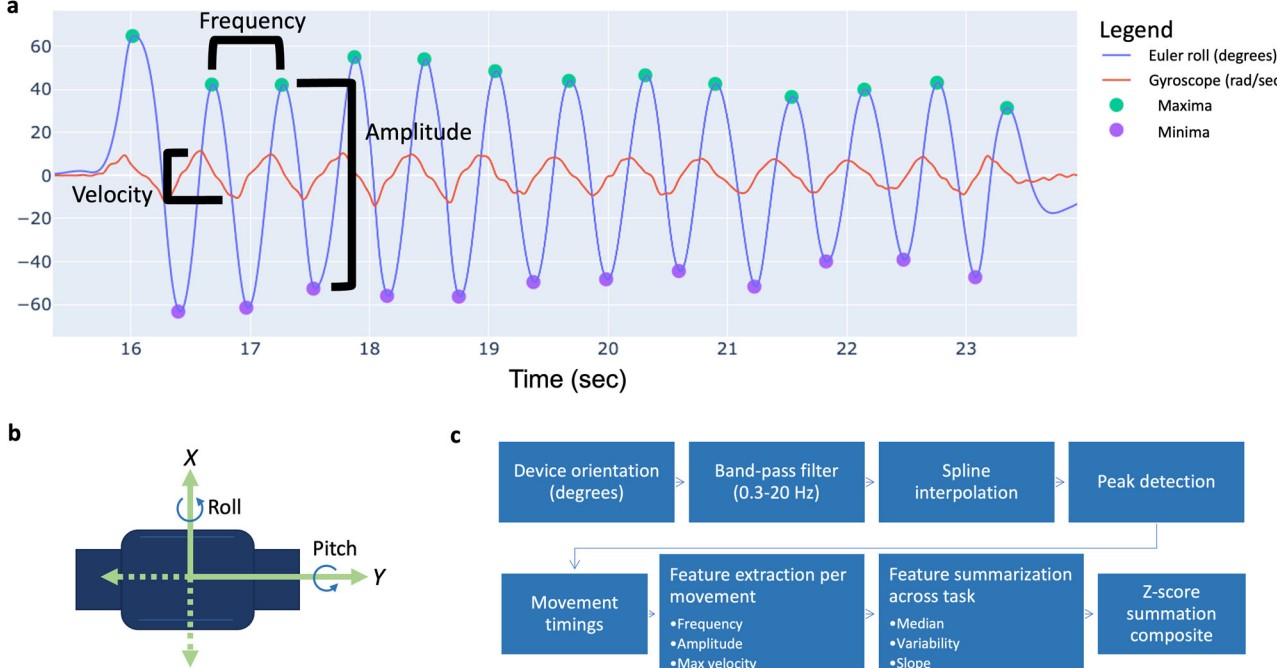

**Fig. 1 | Digital feature derivation from a pronation-supination assessment.**
**a** Euler angle and gyroscope data derived from a sensor-fusion based approach[43] over several seconds is presented during a pronation-supination assessment. **b** Sensor axes in the direction of movement for pronation-supination and toe-tapping assessments were used for data processing. **c** Euler angle data from the appropriate axis for each assessment was algorithmically processed to identify start and end timings of each pronation-supination or toe-tapping movement. Frequency, amplitude, and velocity were derived for each movement and summarized across each task using median, variability, and slope, producing a total of 9 digital features per assessment. Finally, unweighted $z$-score summation was performed to derive a single digital composite score per assessment.

Next, for each assessment, $z$-scores for slope and mean features are multiplied by $-1$ to produce the opposite value and ensure that all $z$-score features increase based on increasing impairment, analogous to MDS-UPDRS ratings. Lastly, the nine $z$-score values are summated for each assessment into a single digital composite score per participant/visit. Of note, the digital composite score is unweighted, in that individual $z$-scores are not weighted or optimized toward enhanced sensitivity in any way prior to summation.

Comparisons of digital measures between baseline and 12-month visits were made using paired two-sided Wilcoxon rank-sum tests, Cohen's $d$ standardized effect size, and the proportion of participants whose score increased from baseline to month 12. Spearman correlation and the Kruskal–Wallis test were used to compare digital measures with MDS-UPDRS clinical scores. Unpaired two-sided Wilcoxon rank-sum tests were used for comparisons between discrete clinical scores and for evaluating known groups validity. Test-retest reliability was assessed using intraclass correlation coefficients (ICC) (two-way random effects, absolute agreement) and Pearson correlations between concurrent visits. Test-retest reliability between visits was assessed according to criteria suggested by Cichetti et al.[36]. Statistical analysis was performed using R v4.2.1[37].

### Reporting summary
Further information on research design is available in the Nature Portfolio Reporting Summary linked to this article.

## Results
### Digital composites demonstrate enhanced sensitivity to disease progression compared to corresponding clinical measurements
In order to investigate how individual digital features contribute to longitudinal disease sensitivity, we compared digital features between baseline and 12 months using paired Wilcoxon signed rank tests in both healthy and PD participants. Out of 9 individual digital features, only slope amplitude and slope maximum velocity demonstrated significant differences between

baseline and 12 months for pronation-supination assessments, though several trended toward significance (Supplementary Table 1). For toe-tapping assessments, only slope frequency demonstrated significant change between baseline and 12 months. In order to combine component features into a single generalized score, we performed unweighted $z$-score summation to derive a pronation-supination digital composite score and a toe-tapping digital composite score. Interestingly, although few individual digital features demonstrate strong sensitivity to longitudinal change, both pronation-supination (Wilcoxon $p = 0.018$, $V = 429$, CI: $-2.76 - -0.27$, effect size $= 0.45$, proportion $= 0.62$) and toe-tapping (Wilcoxon $p = 0.011$, $V = 485$, CI: $-1.83 - -0.23$, effect size $= 0.30$, proportion $= 0.66$) composite scores demonstrated significant differences between baseline and 12 months (Fig. 2a, c). Thus, composite summarization of features may better capture the heterogeneous manifestation of bradykinesia, leading to an additive effect of individual features that increases sensitivity to change.

Next, we compared digital composite scores for both pronation-supination and toe-tapping to relevant scores from the MDS-UPDRS. As stated previously, digital composite scores significantly distinguished between baseline and 12 months in PD patients (Fig. 2a, c). In contrast, neither MDS-UPDRS pronation-supination score (Wilcoxon $p = 0.919$, $V = 193.5$, CI: $-1.00 - 0.50$, effect size $= 0.00$, proportion $= 0.29$), toe-tapping score (Wilcoxon $p = 0.622$, $V = 254.5$, CI: $-1.00 - 0.00$, effect size $= 0.08$, proportion $= 0.36$), or total bradykinesia score (Wilcoxon $p = 0.178$, $V = 398$, CI: $-2.50 - 0.50$, effect size $= 0.10$, proportion $= 0.62$) demonstrated significant differences between baseline and 12 months in PD or healthy comparison participants (Fig. 2b, d and Supplementary Fig. 1b). MDS-UPDRS total part 3 score trended toward significance (Wilcoxon $p = 0.087$, $V = 501$, CI: $-4.00 - 0.50$, effect size $= 0.15$, proportion $= 0.63$) and significant differences between baseline and 12 months were seen with MDS-UPDRS total score (Wilcoxon $p = 0.036$, $V = 420$, CI: $-6.50 - 0.50$, effect size $= 0.21$, proportion $= 0.62$) (Supplementary Fig. 1d).

In healthy participants, the digital pronation-supination composite score did not demonstrate significant change (Wilcoxon $p = 0.904$, $V = 307$,

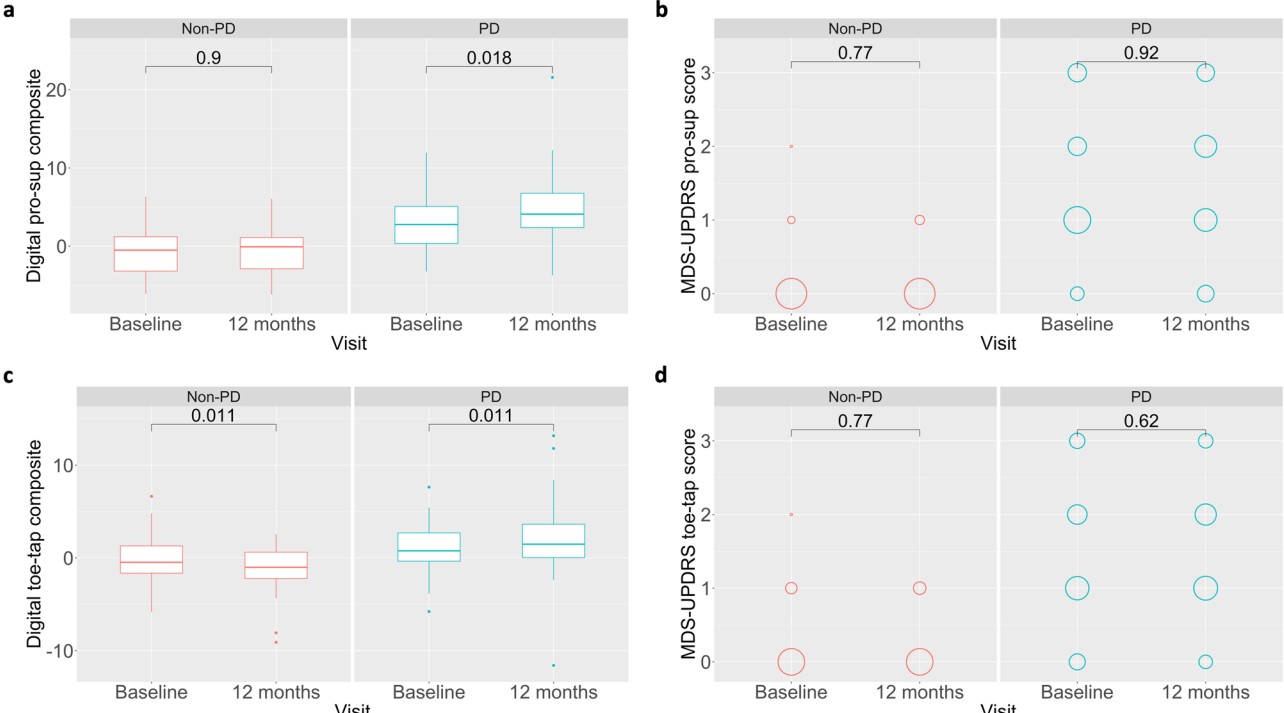

**Fig. 2 | Enhanced sensitivity of digital composites to 1-year longitudinal progression compared to UPDRS pronation-supination score.** a Digital pronation-supination composite scores significantly distinguished between baseline and 12-month visits in PD patients ($n = 52$), whereas no difference was seen in healthy participants ($n = 35$). **b** In contrast, MDS-UPDRS pronation-supination scores demonstrated no difference between baseline and 12-month visits in healthy or PD participants. **c** Similarly, digital toe-tapping composite scores significantly distinguished between baseline and 12-month visits in PD patients ($n = 56$), suggesting worsening performance, and significantly distinguished healthy participants ($n = 35$) in the opposite direction, suggesting improved performance. **d** MDS-UPDRS toe-tapping scores demonstrated no difference between baseline and 12-month visits in healthy or PD participants.

CI: −0.96–0.93, effect size = 0.00, proportion = 0.54) between baseline and 12 months (Fig. 2a). However, surprisingly, the digital toe-tapping composite did significantly distinguish between baseline and 12 months (Wilcoxon $p = 0.011$, $V = 468$, CI: 0.29–2.10, effect size = 0.43, proportion = 0.26), albeit, in the opposite direction as PD patients (Fig. 2c), suggesting improved performance over the course of 12 months in healthy relative to PD participants. It may be the case that the improved performance in healthy participants between baseline and 1-year visits demonstrates a learning effect, which is plausible given healthy volunteers generally lack previous experience conducting bradykinesia-related assessments whereas PD patients may have completed these assessments as part of clinical care or participation in research studies. Our results demonstrate that bradykinetic symptom progression measured via pronation-supination and toe-tapping movements is detectable over 1 year in early PD using digital composite scores whereas such progression is not observed on MDS-UPDRS Part 3 total bradykinesia or item scores.

To evaluate how much time is needed to measure a change in bradykinesia severity, we evaluated whether digital composites or MDS-UPDRS scores were able to distinguish between timepoints less than 12 months. The pronation-supination (Wilcoxon $p = 0.060$, $V = 545$, CI: −2.08–0.04, effect size = 0.27, proportion = 0.60) and toe-tapping (Wilcoxon $p = 0.239$, $V = 653$, CI: −1.48–0.39, effect size = 0.19, proportion = 0.57) digital composites trend to some extent toward distinguishing between baseline and 9 months. Pronation-supination (Wilcoxon $p = 0.962$, $V = 277.5$, CI: −0.50–0.50, effect size = 0.02, proportion = 0.33) and toe-tapping (Wilcoxon $p = 0.403$, $V = 251.5$, CI: −1.00–0.00, effect size = 0.11, proportion = 0.34) MDS-UPDRS scores do not distinguish between baseline and 9 months. Neither digital composites nor MDS-UPDRS scores demonstrate significant changes between baseline and 6 months or baseline and 3 months. Based on our results, a minimum of 12 months was required to measure strong change in bradykinesia severity using digital composites in our study population.

Of the 52 PD participants in the pronation-supination analysis and 56 PD participants in the toe-tapping analysis, 14 matriculated to a symptom managing medication at some point over the course of the 12-month study period. To evaluate the sensitivity of the digital composites in absence of symptomatic medications, change from baseline to the 12-month visit was investigated following removal of 14 patients who commenced medication at any point during the study. Distinguishability between baseline and 12 months increased for both pronation-supination (Wilcoxon $p = 0.006$, $V = 161$, CI: −3.55–−0.57, effect size = 0.62, proportion = 0.67) and toe tapping (Wilcoxon $p = 0.007$, $V = 186$, CI: −2.28–−0.44, effect size = 0.35, proportion = 0.66) digital composites as well as pronation-supination (Wilcoxon $p = 0.488$, $V = 78.5$, CI: −1.00–0.50, effect size = 0.12, proportion = 0.33) and toe-tapping (Wilcoxon $p = 0.046$, $V = 94$, CI: −1.00–0.00, effect size = 0.34, proportion = 0.5) MDS-UPDRS sub-scores.

The overall increase in sensitivity of digital and clinical measures suggest that bradykinesia symptoms are to some degree masked in the presence of a symptomatic treatment, which is consistent with previous work using standard clinical measures[14,38]. It is noteworthy, in the context of real-world settings where patients may begin or continue to take symptom managing medication in parallel with interventions that may slow the course of PD, that digital measurement tools are sensitive to change in function even in the presence of symptomatic treatments. To this end, in a typical care setting it is important to understand progressive changes in function even when symptoms appear to be controlled by standard of care symptomatic treatments such as carbidopa/levodopa.

**Digital composites demonstrate convergent validity with MDS-UPDRS item scores and subscores**
To evaluate the validity of pronation-supination and toe-tapping digital composite scores in relation to the same constructs measured via the MDS-UPDRS, we investigated the relationship between digital composites and corresponding MDS-UPDRS items and sub-scores. The digital

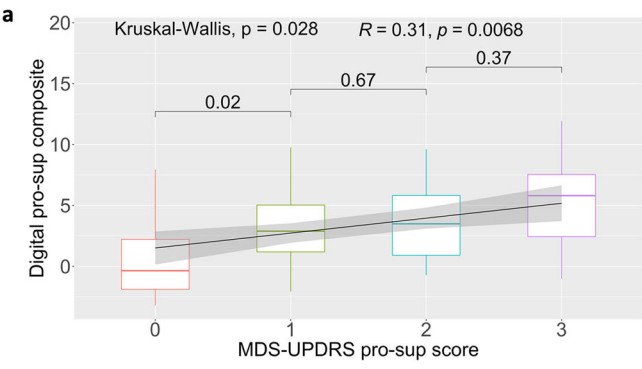

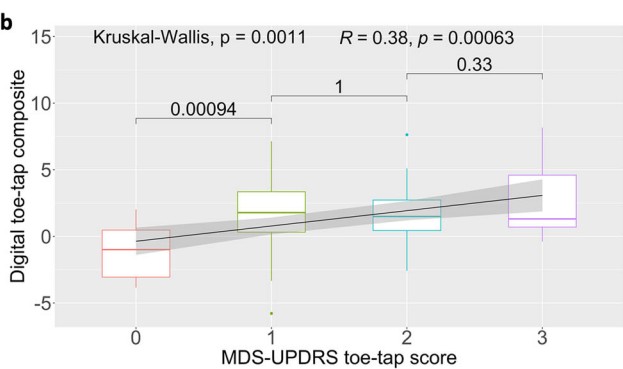

**Fig. 3 | Convergent validity of digital composites.** Digital composite scores vary significantly with MDS-UPDRS pronation-supination (Kruskal–Wallis $p = 0.028$, $n = 76$) and toe-tapping (Kruskal–Wallis $p = 0.001$, $n = 78$) scores at the baseline visit. Both (**a**) pronation-supination (Wilcoxon $p = 0.02$) and (**b**) toe tapping (Wilcoxon $p < 0.001$) digital composite scores significantly distinguished between scores of 0 and 1, though, did not differentiate between scores of 1 and 2, or 2 and 3 (sample sizes and summary metrics for each individual clinical score is available in Supplementary Data).

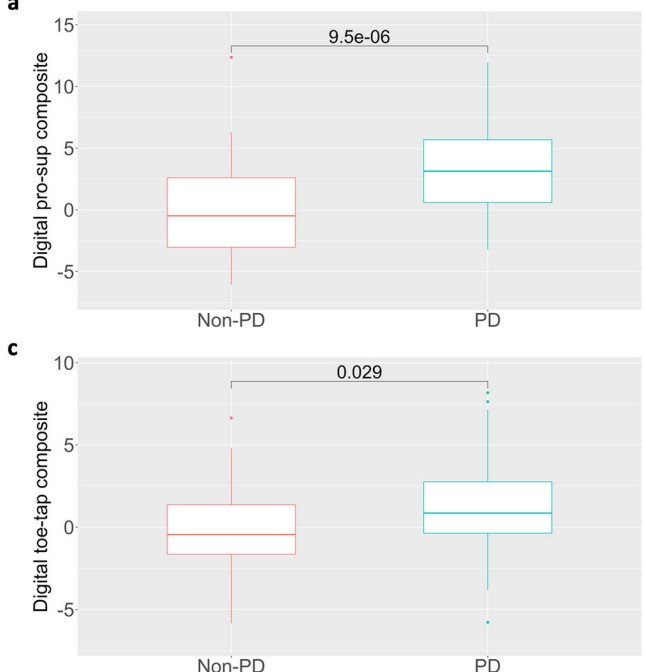

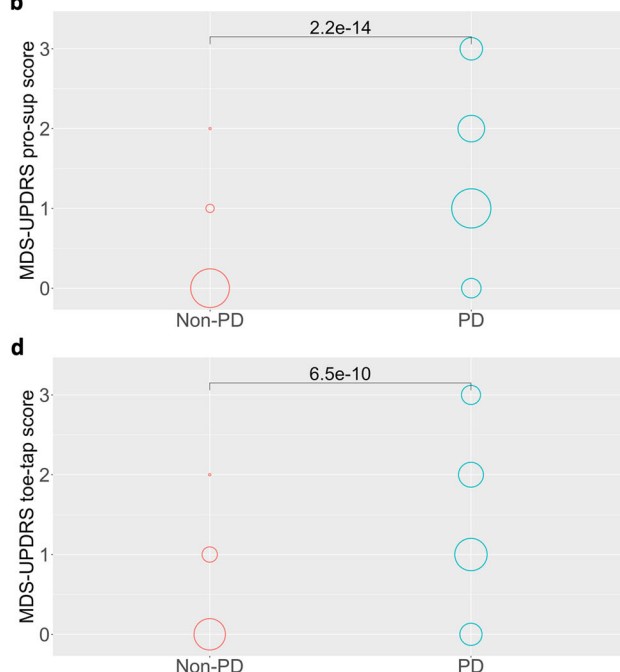

**Fig. 4 | Known groups validity of digital composites.** Both (**a, b**) digital composite scores (Wilcoxon $p < 0.001$) and MDS-UPDRS pronation-supination scores (Wilcoxon $p < 0.001$) significantly distinguished between healthy ($n = 40$) and PD ($n = 76$) participants. Likewise, both (**c, d**) digital composite scores (Wilcoxon $p = 0.029$) and MDS-UPDRS toe-tapping scores (Wilcoxon $p < 0.001$) significantly distinguished between healthy ($n = 39$) and PD ($n = 78$) participants.

pronation-supination composite score demonstrated a significant relationship with MDS-UPDRS pronation-supination item score (item 3.6; Kruskal–Wallis $p = 0.028$; Spearman's rho = 0.31, $p = 0.007$) (Fig. 3a). The digital pronation-supination composite score significantly distinguished between scores of 0 and 1 (Wilcoxon $p = 0.02$), but not between scores of 1 and 2, or 2 and 3. Similarly, the digital toe-tapping composite score showed a significant relationship with MDS-UPDRS toe-tapping item score (item 3.7; Kruskal–Wallis $p = 0.001$; Spearman's rho = 0.38, $p < 0.001$) (Fig. 3b). The digital toe-tapping composite score significantly distinguished between scores of 0 and 1 (Wilcoxon $p < 0.001$), but not between scores of 1 and 2, or 2 and 3. Our results demonstrate evidence to support a good relationship between digital composite scores and their corresponding clinical constructs.

**Digital composites demonstrate known groups validity**
To further investigate the validity of digital composites for pronation-supination and toe-tapping assessments, we evaluate the ability of these measures to differentiate between PD and healthy comparison participants at Baseline. Both pronation-supination (Wilcoxon $p < 0.001$, $W = 757$, CI: $-4.86$–$-1.96$) and toe-tapping (Wilcoxon $p = 0.029$, $W = 1143$, CI: $-2.23$–$-0.13$) digital composites demonstrate significant differences between PD and non-PD participants (Fig. 4a, c). Likewise, both MDS-UPDRS pronation-supination (Wilcoxon $p < 0.001$, $W = 278$, CI: $-1.00$–$-1.00$) and toe-tapping (Wilcoxon $p < 0.001$, $W = 510$, CI: $-1.00$–$-1.00$) sub-scores significantly distinguished between PD and non-PD participants (Fig. 4b, d). In addition to digital composites, the majority of individual digital features significantly distinguished between PD and non-PD participants for both pronation-supination and toe-tapping assessments (Supplementary Table 2).

We also conducted an ROC-AUC sensitivity analysis to investigate the ability of digital composites and MDS-UPDRS scores to differentiate PD from non-PD patients at baseline (Supplementary Fig. 2). Digital composite scores distinguished between PD and non-PD participants for both pronation-supination (AUC = 0.751) and toe-tapping (AUC = 0.624), as

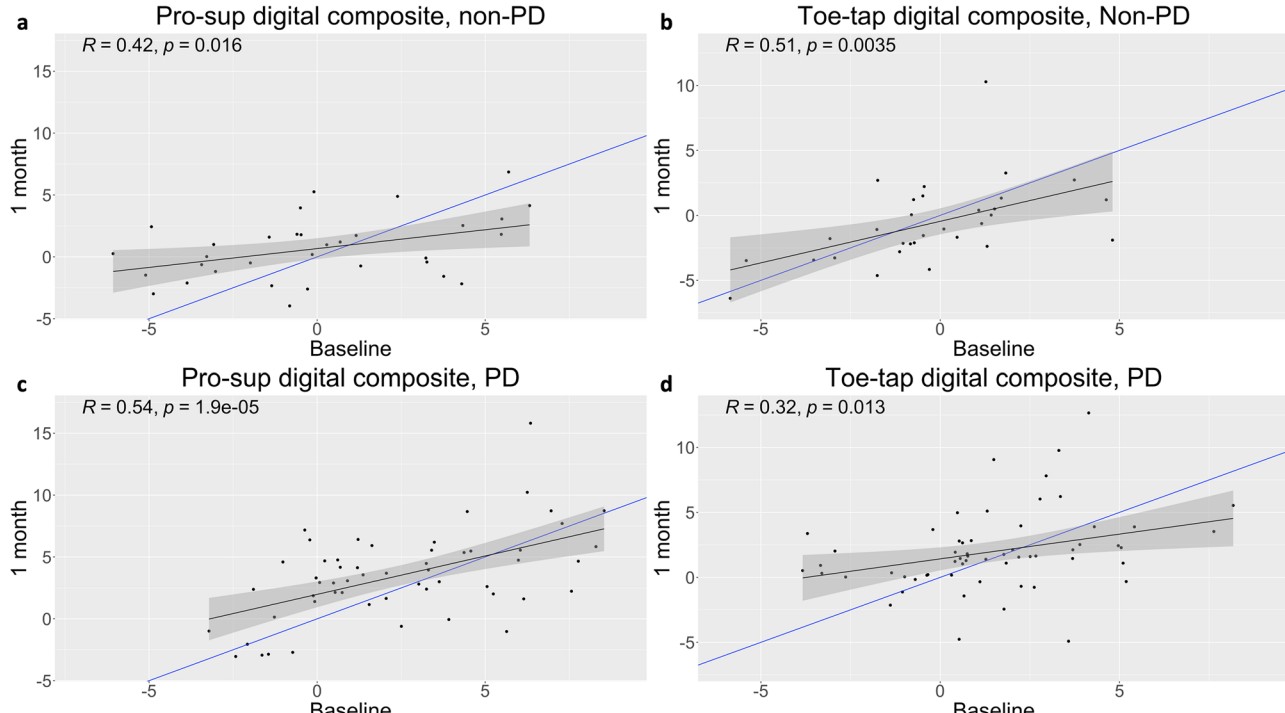

**Fig. 5 | A moderate relationship exists between digital composite values derived between baseline and 1-month visits.** Represented with black lines, significant linear relationships existed ($p < 0.016$) and Pearson $R$ coefficients varied between 0.32 and 0.54 across (**a, b**) pronation-supination assessments and (**c, d**) toe-tapping assessments in both PD and healthy participants (**a**: $n = 33$; **b**: $n = 31$; **c**: $n = 55$; **d**: $n = 59$). For reference, blue lines demonstrate a slope of 1.

did MDS-UPDRS pronation-supination (AUC = 0.910) and toe-tapping (AUC = 0.832) scores. Based on our results, whereas digital composites showed increased sensitivity to change in function over time, MDS-UPDRS item scores showed better ability to differentiate PD from non-PD participants at baseline. One explanation that MDS-UPDRS bradykinesia scores demonstrate strong distinguishability between PD and non-PD may be that bradykinesia is a cardinal motor sign and part of primary differential diagnostic criteria (alongside resting tremor and rigidity) according to UK Brain Bank standards. Thus, since PD participants in this study had a confirmed diagnosis of PD and may have been up to 2 years past initial diagnosis, it would be expected that bradykinesia scores on the MDS-UPDRS, which may also be used during diagnosis, showed enhanced ability to differentiate between the two groups in this study. However, given that bradykinesia may emerge up to a decade prior to a formal diagnosis of PD[39], and our results showing increased sensitivity of digital bradykinesia measures to progression, these digital measures may similarly provide enhanced sensitivity to the initial detection of PD-related bradykinesia prior to clinical diagnosis. Further research should be conducted to investigate digital measure capability in a diagnostic capacity.

**Digital composites demonstrate moderate test-retest reliability**
In order to evaluate test-retest reliability of digital features, Pearson $R$ values and intraclass correlation coefficients were calculated between baseline and 1-month visits in both healthy and PD participants. Follow-up visits at 1 month were chosen to both decrease repeat testing burden within a single clinic visit and to align more closely with previous multisite studies designed to examine test-retest reliability of the MDS-UPDRS in early PD patients[40]. In healthy participants, digital composite scores for both pronation-supination (Pearson $R = 0.42$, $p = 0.016$; ICC = 0.40) and toe-tapping (Pearson $R = 0.51$, $p = 0.004$; ICC = 0.5) demonstrated moderate test-retest reliability between baseline and 1-month visits (Fig. 5a, b and Supplementary Table 3). In PD participants, digital composite scores for pronation-supination (Pearson $R = 0.54$, $p < 0.001$; ICC = 0.52) and toe-tapping (Pearson $R = 0.32$, $p = 0.013$; ICC = 0.31) demonstrated moderate and poor test-retest reliability, respectively, between baseline and 1-month

visits (Fig. 5c, d and Supplementary Table 4). As indicated by the linear regression line (in black) relative to a slope of 1 (in blue), across both healthy and PD participants, poorer performance at baseline resulted in relative improvement at 1 month, and better performance at baseline resulted in relative worsening at 1 month (Fig. 5).

Interestingly, pronation-supination digital composite test-retest reliability improved for both healthy and PD participants during later assessments when comparing between 9-month and 12-month visits, demonstrating excellent reliability in healthy participants (ICC = 0.75), and good reliability in PD participants (ICC = 0.69) (Supplementary Tables 3 and 4). Improved pronation-supination test-retest reliability later in the study may have been due to participants increased familiarity with performance of the assessment over the length of the study. However, this was not consistently the case for toe-tapping, as digital composite test-retest reliability between 9-month and 12-month visits improved from poor to moderate (ICC = 0.5) in PD participants but worsened from moderate to poor in healthy participants (ICC = 0.3) (Supplementary Tables 3 and 4).

**Discussion**
Development of objective clinical measurements to assess the longitudinal status of disease is important for clinical research and therapeutic development, especially in PD where few objective biochemical, genetic, or imaging biomarkers exist. Our results demonstrate that digital composite measures for upper and lower extremity bradykinesia are more sensitive to 1-year longitudinal disease progression in early PD than corresponding MDS-UPDRS items (Fig. 2 and Supplementary Fig. 1). This provides preliminary evidence for enhanced sensitivity of digital measures for monitoring change in PD motor signs over time relative to current measurement standards. Digital measures were also able to differentiate between PD and healthy comparison participants and showed convergent validity with corresponding MDS-UPDRS items and sub-scores. Moderate test-retest reliability was seen between baseline and 1-month visits. From a clinical trial perspective, better measures of progression are needed to optimize trial designs and decrease the duration needed to detect signals of efficacy in proof-of-concept trials of novel therapies. The current results represent a

step toward better long-term characterization of disease course and response to treatment in PD patients that may ultimately lead to enhanced therapeutic trials and care.

Our results support the use of digital health technologies in providing a more sensitive evaluation of bradykinesia than the corresponding MDS-UPDRS items. The ability of objective IMU-based measures to quantify more subtle changes in bradykinesia versus clinical examination likely led to the enhanced sensitivity observed in our study. However, in our opinion, it is likely that sensitivity could be improved further. Our study focused on data collected during periodic in-clinic assessments that may not capture day to day fluctuation in bradykinesia. One promise of DHTs is the ability to conduct more frequent remote assessments or passive monitoring in at-home settings to capture fluctuations and provide a more accurate snapshot of function at a given point in time. It is likely that sensitivity could be increased further if measurements were taken more frequently in home environments, which is feasible given the minimal sensor requirements for upper and lower extremity bradykinesia measurement presented here. Furthermore, composite scores developed for this study were unweighted, and it may be feasible to apply various optimization methodologies to further enhance sensitivity to changes in bradykinesia over time.

Our results also suggest that composite scoring may be an effective method to enhance sensitivity to the progression of bradykinesia or other factors, likely due to heterogeneity in the way bradykinesia may manifest across individuals during structured tasks. The demonstrated composite approach attempts to better align sensor-based measures with how bradykinesia is clinically assessed, namely by examining multiple features, including amplitude, speed, and decrement over time of repetitive movements. Interestingly, unweighted $z$-score digital composites that included all 9 algorithmically derived features demonstrated significant change over 1 year in PD patients, however, the majority of individual digital features for pronation-supination (7/9) and toe-tapping (8/9) did not demonstrate significant change over 1 year. Thus, our results provide evidence that composite scoring may enhance sensitivity to progression by combining various individual features that do not individually change significantly over time but likely trend in a common direction. It is also possible that individual features have reduced statistical power compared to a composite score given that characteristics of bradykinesia may be heterogenous across PD patients. For instance, the pathophysiology of bradykinesia encompasses multiple factors, including muscle weakness, rigidity, tremor, movement variability, and slowing of thought, and depending on an individual's unique disease progression, the characteristics of bradykinesia exhibited may vary[41]. Therefore, digital measurement composite scoring methodologies may enable increased generalizability of motor impairment evaluation in heterogenous patient populations, and may be broadly applicable to other DHT-derived measures in this population.

Furthermore, signal processing-based approaches and composite scoring may also facilitate interpretable, generalizable digital measurements that can be more readily understood in therapeutic trials or care. The algorithm presented in this work derives 9 digital features related to clinical characteristics of bradykinesia, including speed, amplitude, rhythm, slowing of movement, and decrementing amplitude, which are described in the MDS-UPDRS (Fig. 1). Interpretability of digital features is important in order for clinicians and researchers to comprehend and utilize results derived from DHTs in a predictable manner[42]. Our results suggest that a small number of heuristically derived signal features that parallel those measured by a clinician during a routine motor examination can be represented objectively by data collected via body worn IMUs during the same examination.

A limitation with the digital measures derived in this study was the moderate to poor test-retest reliability seen with the digital bradykinesia composite scores between baseline and 1-month visits (Fig. 5 and Supplementary Tables 3 and 4). Despite these results, test-retest reliability of the digital composites improved by the end of the study in PD patients with good reliability (ICC = 0.69) for pronation-supination and moderate

reliability (ICC = 0.49) for toe-tapping assessments. Pronation-supination test-retest reliability also improved to excellent (ICC = 0.75) in healthy participants, however, worsened to poor for toe-tapping (ICC = 0.3). It is possible participants felt more comfortable with the assessments toward the end of the study leading to more stable performance during clinician-led elicitations, thus reducing variability between assessments. However, improvements in assessment standardization may further improve test-retest reliability in future studies. Because recordings were made contemporaneously with clinician ratings, the number of movements per assessment used for analysis varied considerably, from 5 to 25 in the case of pronation-supination. Future efforts to standardize assessments may lead to improved test-retest reliability, including tactics such as completing elicitations for standardized periods of time (e.g., 5 s), increasing the number of assessments over shorter periods of time, or more explicitly standardizing sensor positioning. To this end, more consistent, standardized assessments conducted in remote settings would seem feasible given the algorithm presented and minimal sensor requirements. Furthermore, minimizing time between assessments may help to mitigate any changes in assessment performance due to disease progression.

Furthermore, a limitation of the digital composites was the inability to distinguish between MDS-UPDRS scores of 1, 2, and 3. The inability to distinguish scores of 1, 2, and 3 may be due to several potential reasons. First, clinical ratings are not necessarily an objective ground truth for disease impairment. For example, clinician ratings are subjective and can vary based on the experience of the clinician, leading to inter- and intrarater variability. Second, the nature of comparing an ordinal with a continuous scale is challenging. For example, a participant that may fall between a score of 1 and 2 on a linear scale, must be grouped into either 1 or 2 in an ordinal scale, creating discrepancy between the two scales. Lastly, sample sizes of each individual clinical score (0, 1, 2, and 3) are relatively low in this study. Specifically, for pronation-supination sample sizes were 9, 38, 17, and 12, and for toe tapping sample sizes were 15, 33, 19, and 11 for scores of 0, 1, 2 and 3, respectively. Thus, it is difficult to make strong conclusions related to the ability of the digital composites to distinguish between clinical scores. However, future studies with larger sample sizes may help to further optimize the algorithm and evaluate construct validity with greater statistical power.

Digital health technologies have great potential for improving the clinical assessment of motor features in early PD, however there is a lack of data demonstrating that measures derived from DHTs are more sensitive to decline in function over time. Our results suggest that digital measures may enhance sensitivity to the progression of bradykinesia in early PD compared to traditional assessments. Future research should extend our findings by evaluating the sensitivity of digital measures to disease progression and treatment effects relative to current clinical assessments across indications and disease stages.

## Data availability
The data that support the findings of this study are not openly available due to data privacy controls in place as part of the consortium that this work support, and all data are available to members of the Critical Path for Parkinson's Consortium 3DT Initiative Stage 2. For those who are not a part of 3DT Stage 2, a proposal may be made to the WATCH-PD Steering Committee (via the corresponding author) for deidentified baseline datasets. We are unable to provide the raw numerical data for Figs. 2–5 due to the requirement that data is restricted to members of the Critical Path for Parkinson's Consortium 3DT Initiative Stage 2. The summary statistics underlying Figs. 2–5 are available in the Supplementary Data.

## Code availability
Pseudocode is available within the Supplementary Information (Supplementary Note). Additional code that is specific to our data pipeline is available from the authors upon request.

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

## Acknowledgements
Funding for the study was contributed by Biogen, Takeda, and the members of the Critical Path for Parkinson's Consortium 3DT Initiative, Stage 2.

## Author contributions
Conception and paper drafting: M.D.C. and J.C. Data processing and statistical analysis: M.D.C. and D.B. Interpretation of data and critical revision of the manuscript: M.D.C., J.C., L.Y., J.S., M.C., T.K., E.R.D. and J.L.A.

## Competing interests
The authors declare the following competing interests: M.D.C., L.Y., J.S., M.C., and J.D.C. are employees of and own stock in AbbVie Pharmaceuticals. J.L.A. has received compensation for consulting services from VisualDx and the Huntington Study Group; and research support from Biogen, Biosensics, Huntington Study Group, Michael J. Fox Foundation, National Institutes of Health/National Institute of Neurological Disorders and Stroke, NeuroNext Network, and Safra Foundation. T.K. is an employee of and owns stock in Takeda Pharmaceuticals, Inc. D.B. was an employee of AbbVie Pharmaceuticals during the time he contributed to this research article. He has no conflicts or interests at the present time. E.R.D. has received compensation for consulting services from Abbott, Abbvie, Acadia, Acorda, Bial-Biotech Investments, Inc., Biogen, Boehringer Ingelheim, California Pacific Medical Center, Caraway Therapeutics, CuraSen Therapeutics, Denali Therapeutics, Eli Lilly, Genentech/Roche, Grand Rounds, Huntington Study Group, Informa Pharma Consulting, Karger Publications, LifeSciences Consultants, MCM Education, Mediflix, Medopad, Medrhythms, Merck, Michael J. Fox Foundation, NACCME, Neurocrine, NeuroDerm, NIH, Novartis, Origent Data Sciences, Otsuka, Physician's Education Resource, Praxis, PRIME Education, Roach, Brown, McCarthy & Gruber, Sanofi, Seminal Healthcare, Spark, Springer Healthcare, Sunovion Pharma, Theravance, Voyager and WebMD; research support from Biosensics, Burroughs Wellcome Fund, CuraSen, Greater Rochester Health Foundation, Huntington Study Group, Michael J. Fox Foundation, National Institutes of Health, Patient-Centered Outcomes Research Institute, Pfizer, Photo-Pharmics, Safra Foundation, and Wave Life Sciences; editorial services for Karger Publications; stock in Included Health and in Mediflix, and ownership interests in SemCap.
