## [Peer Review File · Communications Medicine]

Reviewers' comments:

Reviewer #1 (Remarks to the Author):

The authors conducted a study to validate a machine-learning algorithm for measuring the severity of Parkinson's disease (PD) using data from wearable sensors placed on both feet and wrists. This data was sourced from 82 early, de-novo PD patients and 50 age-matched controls. The authors demonstrated that the digital composite effectively identified changes from baseline to the 1-year follow-up in PD patients but not in the control group. Moreover, the digital composite for toe-tap and pronation-supination exercises successfully differentiated between PD and control groups. Finally, the digital composite for pronation-supination and toe-tap significantly distinguished between scores 0 and 1 in the relevant MDS-UPDRS items, but it did not differentiate among scores 1, 2, and 3. Consequently, the authors conclude that digital devices might be instrumental in monitoring the progression severity of PD.

This study addresses a significant clinical issue and presents a potential method to monitor longitudinal changes in PD severity, effectively tracking disease progression. The research design is robust, with a commendable sample size and suitable controls. Additionally, the manuscript is coherently written. However, I have a few suggestions to enhance the manuscript:

1. The authors mentioned recruiting untreated PD patients. I am curious if any of these patients commenced medication during the study. If they did, were such participants excluded from the final analysis?
2. Including the ROC curve along with sensitivity and accuracy metrics when differentiating between PD and control would offer more depth to the results.
3. The inability of the digital composite to distinguish among scores 1, 2, and 3 in the MDS-UPDRS is a notable limitation. Elaborating on the potential reasons behind this and discussing possible solutions or future improvements to address this limitation would strengthen the paper.

Reviewer #2 (Remarks to the Author):

This manuscript develops a wearable sensor-based digital algorithm for deriving features of upper and lower-body bradykinesia. Clinical validation is performed by investigating convergent validity with appropriate clinical constructs, known groups validity, and test-retest reliability. However, the present quality of this paper can not be accepted for publication due to the following questions:

1. What is digital composite? How the digital composite score was defined or calculated?
2. What's the contribution of this study? The novelty should be state clearly.
3. To what degree the sensitivity was improved by the proposed score?
4. The authors displayed the comparison of "baseline VS 12 months" were compared, how about baseline VS 3 months, 6 months, and 9 months?

5. The digital score and the clinical measurement may describe the same thing, they may not independent. In the reviewer's view, the digital score is the objectification and quantification of the MDS-UYPDRS score. Therefore, can Kruskal-Wallis test be applied for comparison?

6. How the test-retest reliability was defined? It looks like in the longitudinal comparison? Since the disease is developing all the time, it's normal that the retest in 1 month varies from the test in baseline. Test-retest may be used to describe the tests of same situation, so it may be confusing in this manuscript.

Reviewer #1 (Remarks to the Author):

The authors conducted a study to validate a machine-learning algorithm for measuring the severity of Parkinson's disease (PD) using data from wearable sensors placed on both feet and wrists. This data was sourced from 82 early, de-novo PD patients and 50 age-matched controls. The authors demonstrated that the digital composite effectively identified changes from baseline to the 1-year follow-up in PD patients but not in the control group. Moreover, the digital composite for toe-tap and pronation-supination exercises successfully differentiated between PD and control groups. Finally, the digital composite for pronation-supination and toe-tap significantly distinguished between scores 0 and 1 in the relevant MDS-UPDRS items, but it did not differentiate among scores 1, 2, and 3. Consequently, the authors conclude that digital devices might be instrumental in monitoring the progression severity of PD.

This study addresses a significant clinical issue and presents a potential method to monitor longitudinal changes in PD severity, effectively tracking disease progression. The research design is robust, with a commendable sample size and suitable controls. Additionally, the manuscript is coherently written. However, I have a few suggestions to enhance the manuscript:

1. The authors mentioned recruiting untreated PD patients. I am curious if any of these patients commenced medication during the study. If they did, were such participants excluded from the final analysis?

Thank you for your important question. Indeed, 14 patients commenced medication during the study. We decided to include these patients in the final analysis because we sought to evaluate digital measurement tools in the context of real-world conditions. For example, a patient starting a disease modifying treatment may likely, in parallel, begin/continue to take symptom managing medication, such as carbidopa/levodopa, in order to maintain the ability to complete activities of daily living and quality of life. Although symptomatic treatment may decrease symptom severity, it is still important in the context of real-world settings to develop and evaluate measurement tools that are sensitive to change in function, even with background symptomatic treatments.

However, we agree that evaluating sensitivity of a measurement tool in controlled settings is also valuable. Therefore, we conducted a sub-analysis, which we have included in the Results section, in which the 14 patients who commenced medication were removed from analysis. We evaluated change from baseline to the 12-month visit using paired, two-sided Wilcoxon rank sum tests. In order to expand the analysis, we also included Cohen's d standardized effect size as well the proportion of participants whose score increased from baseline to month 12. Consistent with previous work (Holden et al. 2018; DOI: 10.1002/mdc3.12553), distinguishability between baseline and 12 months increased for both the digital composites and MDS-UPDRS sub-scores. The overall increase in sensitivity of both digital and clinical measures suggests that, indeed, progression of bradykinesia is to some degree masked in the presence of a symptomatic treatment. The Results section (final 2 paragraphs of the section titled 'Digital composites demonstrate enhanced sensitivity to disease progression compared to corresponding clinical measurements') and Methods section have been adjusted to include our analysis and results.

2. Including the ROC curve along with sensitivity and accuracy metrics when differentiating between PD and control would offer more depth to the results.

Thank you for your suggestion. Our intent for the known groups analysis was to demonstrate clinical validity of the digital measures, as the focus of the manuscript was longitudinal change of function and not necessarily diagnostic capability. However, we have included an AUC and ROC curve analysis in the Supplementary section to support the known groups validity analysis. Based on AUC values, digital composites for both pronation-supination (AUC=0.751) and toe-tapping (AUC=0.624) show moderate diagnostic discrimination between PD and non-PD participants. MDS-UPDRS pronation-supination (AUC=0.910) and toe-tapping (AUC=0.832) scores show good diagnostic sensitivity. Based on our results, as opposed to increased sensitivity to change in function over time, digital composites computed in this study do not show enhanced ability to differentiate healthy volunteers from patients who have already received a confirmed diagnosis of PD. One explanation for strong distinguishability between PD and non-PD may be that, according to UK Brain Bank standards, bradykinesia is part of the primary differential diagnostic criteria, alongside resting tremor and rigidity. Thus, since PD participants in this study were within 2 years of an initial diagnosis of PD, it is expected that MDS-UPDRS bradykinesia scores, which may also be used during diagnosis, showed an enhanced ability to distinguish from non-PD participants. However, given that bradykinesia may emerge up to a decade prior to a formal diagnosis of PD (Postuma et al. 2012; DOI: 10.1093/brain/aws093), and our results showing increased sensitivity of digital bradykinesia measures to progression, these digital measures may similarly provide enhanced sensitivity to the initial detection of PD-related bradykinesia prior to clinical diagnosis. We believe further research should be conducted to investigate digital measure capability in a diagnostic capacity. We have added text to the Results section (final paragraph of the section titled ‘Digital composites demonstrate known groups validity’) to reflect our results and interpretation.

3. The inability of the digital composite to distinguish among scores 1, 2, and 3 in the MDS-UPDRS is a notable limitation. Elaborating on the potential reasons behind this and discussing possible solutions or future improvements to address this limitation would strengthen the paper.

Thank you for your suggestion. The inability of the digital composites to distinguish between scores of 1, 2, and 3 in the MDS-UPDRS may be due to several reasons. First, clinical ratings are not necessarily an objective ground truth for disease impairment. For example, clinician ratings are subjective and can vary based on the experience of the clinician, leading to inter- and intrarater variability. Second, the nature of comparing an ordinal with a continuous scale is challenging. For example, a participant that may fall between a score of 1 and 2 on a linear scale, must be grouped into either 1 or 2 in an ordinal scale, creating discrepancy between the two scales. Lastly, sample sizes of each individual clinical score (0, 1, 2, and 3) are relatively low in this study. Specifically, for pronation-supination sample sizes were 9, 38, 17, and 12, and for toe tapping sample sizes were 15, 33, 19, and 11 for scores of 0, 1, 2 and 3, respectively. Thus, it is difficult to make strong conclusions related to the ability of the digital composites to distinguish between clinical scores. However, future studies with larger sample sizes may help to further

investigate this limitation. We have added text to the Discussion section (6th paragraph) to reflect our viewpoint.

Reviewer #2 (Remarks to the Author):

This manuscript develops a wearable sensor-based digital algorithm for deriving features of upper and lower-body bradykinesia. Clinical validation is performed by investigating convergent validity with appropriate clinical constructs, known groups validity, and test-retest reliability. However, the present quality of this paper can not be accepted for publication due to the following questions:

1. What is digital composite? How the digital composite score was defined or calculated?

Thank you for your question. The digital composite refers to a composite summary metric of 9 individual features estimated for each pronation-supination or toe-tapping assessment. Specifically, 3 features (frequency, amplitude, and max velocity) are estimated per pronation-supination and toe-tapping movement. Next, 3 summary metrics (median, standard deviation, and slope) are used to summarize each individual movement into a single measurement per assessment (ie. Mean frequency, standard deviation of frequency, slope of frequency). This summarization results in 9 features per task (supplementary table 1) which are combined to a single digital composite score, estimated using normalized z-score summation. To calculate the z-score digital composite, z-scores for each of the 9 features per assessment are estimated by normalizing by the corresponding feature in non-PD controls from the baseline visit. For example, in the case of slope frequency, slope frequency for each assessment is subtracted by the mean of the slope frequency in non-PD controls at baseline and the resulting value is divided by the standard deviation of slope frequency in non-PD controls at baseline.

$$Z \text{ score} = (\text{feature} - \text{mean}(\text{feature in non-PD at baseline})) / \text{sd}(\text{feature in non-PD at baseline})$$

Next, for each assessment, z-scores for slope and mean features are multiplied by -1 to produce the opposite value and ensure that all z-score features increase based on increasing impairment, analogous to MDS-UPDRS ratings. Lastly, the 9 z-score values are summated for each assessment into a single digital composite score. Of note, we also refer to this score as unweighted, in that individual z-scores are not weighted or optimized toward enhanced sensitivity in any way prior to summation. We agree that detailed explanation of digital composite derivation was lacking in the manuscript. To this end, we have added clarifying text to the Methods section in line with what is presented above.

2. What's the contribution of this study? The novelty should be state clearly.

Thank you for asking for clarification. Digital health is an emerging field that has wide implications in the field of clinical measurement and healthcare as a whole. A large number of studies have investigated digital health technologies in the context of accuracy and clinical validity, demonstrating that indeed wearable devices are able to measure movement and disease

symptoms accurately and, in many cases, comparably to expert clinician raters. Thus, there has been much discussion surrounding the potential of digital technologies to increase sensitivity of clinical measurement, however, sparse data to support the actual benefit relative to current clinical standards. This is especially true in the context of monitoring PD progression, where little information currently exists regarding the ability of sensor-based measures to track progressive motor changes in PD patients. From a clinical trial perspective, better measures of progression are needed to optimize trial designs and decrease the duration needed to detect signals of efficacy in proof-of-concept trials of novel therapies, and our results represent a step toward better long-term characterization of disease course and response to treatment in PD patients. Thus, the findings and perspective presented in this study are novel and important, as we directly evaluate the ability of digital technologies to measure patient function over time compared to current clinical standards (i.e., MDS-UPDRS). The results of the study demonstrate the power of digital approaches to enhance measurement sensitivity to longitudinal disease progression.

Furthermore, there are questions surrounding how to effectively translate wearable sensor data into clinically meaningful information. Investigation of summarization and composite scoring methodologies that bring sensor-based measure into alignment with how patients are rated clinically, such as the normalized z-score digital composite presented here, are lacking in the digital health field. The findings in our study are unique in that we demonstrate the power of a composite scoring approach that is more closely aligned with how clinicians typically assess bradykinesia versus previous work examining different features in isolation or that do not directly relate to clinical assessment of PD.

Lastly, many questions remain related to when and how digital health technologies can be used to enhance measurement sensitivity and generate clinical insight. However, publicly available algorithms for processing data are limited, especially for task-based assessment of bradykinesia. To this end, we describe our algorithm methodology and include pseudocode so that other researchers may adapt, improve, and test our methodology on other datasets. We have added clarifying text to the Discussion section (first and third paragraphs) of the manuscript reflecting our viewpoint.

3. To what degree the sensitivity was improved by the proposed score?

Thank you for your important question. We chose to evaluate the differences in sensitivity using paired two-sided Wilcoxon rank-sum tests. We believe this analysis methodology is useful for determining whether there is a statistical difference between the two time points evaluated in the study, and thus a change in symptom impairment between baseline and 12 months. Although it is difficult to make a direct comparison of the degree of sensitivity between the MDS-UPDRS, a short ordinal scale ranging between 0 and 4, and the continuous digital composite scores, ranging from approximately -6 to 21, we believe that the use of standardized effect size and the proportion of participants whose score increased from baseline to month 12 help to clarify the degree of change.

Therefore, in order to provide further information related to the degree of change between the two visits, we have added statistical details to the manuscript, including Wilcoxon test statistics, confidence intervals, Cohen's d standardized effect sizes, and the proportion of participants whose score increased from baseline to month 12. We have adjusted the Results and Methods sections to include these analysis methods.

4. The authors displayed the comparison of “baseline VS 12 months” were compared, how about baseline VS 3 months, 6 months, and 9 months?

Thank you for your question. Baseline versus 12 months was the focus of the manuscript as 12 months was the minimum amount of time to measure strong changes in bradykinesia severity using digital measures in our study population. However, we have added the results of a sub-analysis comparing baseline versus 9, 6, and 3 months within the Results section (4th paragraph of the 1st section). The digital composites trend toward distinguishing between baseline and 9 months, though 12 months is required to measure statistically significant change.

5. The digital score and the clinical measurement may describe the same thing, they may not independent. In the reviewer's view, the digital score is the objectification and quantification of the MDS-UYPDRS score. Therefore, can Kruskal-Wallis test be applied for comparison?

Thank you for your question. Although the digital score and clinical measures are both describing the phenomenon of bradykinesia impairment, they are independent measures. For example, when assigning an MDS-UPDRS score, clinicians do not have access to digital measures, and their ratings are solely based on their expert judgement. Likewise, clinician ratings are not used to derive or influence the values of digital measures. Thus, both MDS-UPDRS scores and digital measures presented here are independent as the occurrence of one does not affect the probability of occurrence of the other. Furthermore, there is some precedence in the literature for using Kruskal-Wallis test as a method to compare between digital measures and clinical scores (Mahadevan et al. 2020; DOI: 10.1038/s41746-019-0217-7). In addition to Kruskal-Wallis, we include Spearman correlation to evaluate the relationship between the two variables as well as unpaired two-sided Wilcoxon rank-sum tests for comparisons between discrete clinical scores. If this explanation is not sufficient, we are glad to consider making changes based on feedback.

6. How the test-retest reliability was defined? It looks like in the longitudinal comparison? Since the disease is developing all the time, it's normal that the retest in 1 month varies from the test in baseline. Test-retest may be used to describe the tests of same situation, so it may be confusing in this manuscript.

Thank you for your questions. Ideally, from a statistical standpoint, test-retest reliability would be conducted on as short a timescale as possible to limit any changes due to disease progression or other extraneous factors that may affect performance. However, test-retest periods of two weeks are typically recommended as the 'standard' cadence of repeat assessments (Streiner et al. 2014; DOI: 10.1111/jan.12402), and in fact the primary study examining the multisite test-retest

reliability of the MDS-UPDRS in early PD patients had a mean test-retest duration of 14.6 days (range 3-36 days; Siderowf et al. 2002; DOI: 10.1002/mds.10011) which is not far from the duration between measures in the current study. In addition, for this study, in which participants must spend several hours at the clinic to perform various procedures related to the MDS-UPDRS, it was not feasible to repeat assessments twice at the same visit. Despite the 1 month difference between visits, in the case of early Parkinson's disease, large changes in disease severity are unlikely to occur over such short timescales, and indeed our own results suggest 12 months is required to see statistical changes in bradykinesia severity. Thus, the 1-month timepoint was used to evaluate test-retest reliability as minimal changes due to disease state would have occurred and we feel the value of including this analysis in the manuscript supersedes the limitations, as major fluctuations at one month would indicate unreliable, spurious measurements, calling into question the validity of the digital measures. We have added text to the Results section (1st paragraph of the section titled 'Digital composites demonstrate moderate test-retest reliability') and the Discussion section (5th paragraph) to clarify our reasoning for the test-retest reliability analysis.

REVIEWERS' COMMENTS:

Reviewer #1 (Remarks to the Author):

I have reviewed the revisions to the manuscript and find that the changes adequately address my previous concerns. The manuscript has improved significantly.

Reviewer #2 (Remarks to the Author):

The issues has been responded by the authors. The revised paper can be accepted for publication.

Reviewer #1 (Remarks to the Author):

The authors conducted a study to validate a machine-learning algorithm for measuring the severity of Parkinson's disease (PD) using data from wearable sensors placed on both feet and wrists. This data was sourced from 82 early, de-novo PD patients and 50 age-matched controls. The authors demonstrated that the digital composite effectively identified changes from baseline to the 1-year follow-up in PD patients but not in the control group. Moreover, the digital composite for toe-tap and pronation-supination exercises successfully differentiated between PD and control groups. Finally, the digital composite for pronation-supination and toe-tap significantly distinguished between scores 0 and 1 in the relevant MDS-UPDRS items, but it did not differentiate among scores 1, 2, and 3. Consequently, the authors conclude that digital devices might be instrumental in monitoring the progression severity of PD.

This study addresses a significant clinical issue and presents a potential method to monitor longitudinal changes in PD severity, effectively tracking disease progression. The research design is robust, with a commendable sample size and suitable controls. Additionally, the manuscript is coherently written. However, I have a few suggestions to enhance the manuscript:

1. The authors mentioned recruiting untreated PD patients. I am curious if any of these patients commenced medication during the study. If they did, were such participants excluded from the final analysis?

Thank you for your important question. Indeed, 14 patients commenced medication during the study. We decided to include these patients in the final analysis because we sought to evaluate digital measurement tools in the context of real-world conditions. For example, a patient starting a disease modifying treatment may likely, in parallel, begin/continue to take symptom managing medication, such as carbidopa/levodopa, in order to maintain the ability to complete activities of daily living and quality of life. Although symptomatic treatment may decrease symptom severity, it is still important in the context of real-world settings to develop and evaluate measurement tools that are sensitive to change in function, even with background symptomatic treatments.

However, we agree that evaluating sensitivity of a measurement tool in controlled settings is also valuable. Therefore, we conducted a sub-analysis, which we have included in the Results section, in which the 14 patients who commenced medication were removed from analysis. We evaluated change from baseline to the 12-month visit using paired, two-sided Wilcoxon rank sum tests. In order to expand the analysis, we also included Cohen's d standardized effect size as well the proportion of participants whose score increased from baseline to month 12. Consistent with previous work (Holden et al. 2018; DOI: 10.1002/mdc3.12553), distinguishability between baseline and 12 months increased for both the digital composites and MDS-UPDRS sub-scores. The overall increase in sensitivity of both digital and clinical measures suggests that, indeed, progression of bradykinesia is to some degree masked in the presence of a symptomatic treatment. The Results section (final 2 paragraphs of the section titled 'Digital composites demonstrate enhanced sensitivity to disease progression compared to corresponding clinical measurements') and Methods section have been adjusted to include our analysis and results.

2. Including the ROC curve along with sensitivity and accuracy metrics when differentiating between PD and control would offer more depth to the results.

Thank you for your suggestion. Our intent for the known groups analysis was to demonstrate clinical validity of the digital measures, as the focus of the manuscript was longitudinal change of function and not necessarily diagnostic capability. However, we have included an AUC and ROC curve analysis in the Supplementary section to support the known groups validity analysis. Based on AUC values, digital composites for both pronation-supination (AUC=0.751) and toe-tapping (AUC=0.624) show moderate diagnostic discrimination between PD and non-PD participants. MDS-UPDRS pronation-supination (AUC=0.910) and toe-tapping (AUC=0.832) scores show good diagnostic sensitivity. Based on our results, as opposed to increased sensitivity to change in function over time, digital composites computed in this study do not show enhanced ability to differentiate healthy volunteers from patients who have already received a confirmed diagnosis of PD. One explanation for strong distinguishability between PD and non-PD may be that, according to UK Brain Bank standards, bradykinesia is part of the primary differential diagnostic criteria, alongside resting tremor and rigidity. Thus, since PD participants in this study were within 2 years of an initial diagnosis of PD, it is expected that MDS-UPDRS bradykinesia scores, which may also be used during diagnosis, showed an enhanced ability to distinguish from non-PD participants. However, given that bradykinesia may emerge up to a decade prior to a formal diagnosis of PD (Postuma et al. 2012; DOI: 10.1093/brain/aws093), and our results showing increased sensitivity of digital bradykinesia measures to progression, these digital measures may similarly provide enhanced sensitivity to the initial detection of PD-related bradykinesia prior to clinical diagnosis. We believe further research should be conducted to investigate digital measure capability in a diagnostic capacity. We have added text to the Results section (final paragraph of the section titled ‘Digital composites demonstrate known groups validity’) to reflect our results and interpretation.

3. The inability of the digital composite to distinguish among scores 1, 2, and 3 in the MDS-UPDRS is a notable limitation. Elaborating on the potential reasons behind this and discussing possible solutions or future improvements to address this limitation would strengthen the paper.

Thank you for your suggestion. The inability of the digital composites to distinguish between scores of 1, 2, and 3 in the MDS-UPDRS may be due to several reasons. First, clinical ratings are not necessarily an objective ground truth for disease impairment. For example, clinician ratings are subjective and can vary based on the experience of the clinician, leading to inter- and intrarater variability. Second, the nature of comparing an ordinal with a continuous scale is challenging. For example, a participant that may fall between a score of 1 and 2 on a linear scale, must be grouped into either 1 or 2 in an ordinal scale, creating discrepancy between the two scales. Lastly, sample sizes of each individual clinical score (0, 1, 2, and 3) are relatively low in this study. Specifically, for pronation-supination sample sizes were 9, 38, 17, and 12, and for toe tapping sample sizes were 15, 33, 19, and 11 for scores of 0, 1, 2 and 3, respectively. Thus, it is difficult to make strong conclusions related to the ability of the digital composites to distinguish between clinical scores. However, future studies with larger sample sizes may help to further

investigate this limitation. We have added text to the Discussion section (6th paragraph) to reflect our viewpoint.

Reviewer #2 (Remarks to the Author):

This manuscript develops a wearable sensor-based digital algorithm for deriving features of upper and lower-body bradykinesia. Clinical validation is performed by investigating convergent validity with appropriate clinical constructs, known groups validity, and test-retest reliability. However, the present quality of this paper can not be accepted for publication due to the following questions:

1. What is digital composite? How the digital composite score was defined or calculated?

Thank you for your question. The digital composite refers to a composite summary metric of 9 individual features estimated for each pronation-supination or toe-tapping assessment. Specifically, 3 features (frequency, amplitude, and max velocity) are estimated per pronation-supination and toe-tapping movement. Next, 3 summary metrics (median, standard deviation, and slope) are used to summarize each individual movement into a single measurement per assessment (ie. Mean frequency, standard deviation of frequency, slope of frequency). This summarization results in 9 features per task (supplementary table 1) which are combined to a single digital composite score, estimated using normalized z-score summation. To calculate the z-score digital composite, z-scores for each of the 9 features per assessment are estimated by normalizing by the corresponding feature in non-PD controls from the baseline visit. For example, in the case of slope frequency, slope frequency for each assessment is subtracted by the mean of the slope frequency in non-PD controls at baseline and the resulting value is divided by the standard deviation of slope frequency in non-PD controls at baseline.

$$Z \text{ score} = (\text{feature} - \text{mean}(\text{feature in non-PD at baseline})) / \text{sd}(\text{feature in non-PD at baseline})$$

Next, for each assessment, z-scores for slope and mean features are multiplied by -1 to produce the opposite value and ensure that all z-score features increase based on increasing impairment, analogous to MDS-UPDRS ratings. Lastly, the 9 z-score values are summated for each assessment into a single digital composite score. Of note, we also refer to this score as unweighted, in that individual z-scores are not weighted or optimized toward enhanced sensitivity in any way prior to summation. We agree that detailed explanation of digital composite derivation was lacking in the manuscript. To this end, we have added clarifying text to the Methods section in line with what is presented above.

2. What's the contribution of this study? The novelty should be state clearly.

Thank you for asking for clarification. Digital health is an emerging field that has wide implications in the field of clinical measurement and healthcare as a whole. A large number of studies have investigated digital health technologies in the context of accuracy and clinical validity, demonstrating that indeed wearable devices are able to measure movement and disease

symptoms accurately and, in many cases, comparably to expert clinician raters. Thus, there has been much discussion surrounding the potential of digital technologies to increase sensitivity of clinical measurement, however, sparse data to support the actual benefit relative to current clinical standards. This is especially true in the context of monitoring PD progression, where little information currently exists regarding the ability of sensor-based measures to track progressive motor changes in PD patients. From a clinical trial perspective, better measures of progression are needed to optimize trial designs and decrease the duration needed to detect signals of efficacy in proof-of-concept trials of novel therapies, and our results represent a step toward better long-term characterization of disease course and response to treatment in PD patients. Thus, the findings and perspective presented in this study are novel and important, as we directly evaluate the ability of digital technologies to measure patient function over time compared to current clinical standards (i.e., MDS-UPDRS). The results of the study demonstrate the power of digital approaches to enhance measurement sensitivity to longitudinal disease progression.

Furthermore, there are questions surrounding how to effectively translate wearable sensor data into clinically meaningful information. Investigation of summarization and composite scoring methodologies that bring sensor-based measure into alignment with how patients are rated clinically, such as the normalized z-score digital composite presented here, are lacking in the digital health field. The findings in our study are unique in that we demonstrate the power of a composite scoring approach that is more closely aligned with how clinicians typically assess bradykinesia versus previous work examining different features in isolation or that do not directly relate to clinical assessment of PD.

Lastly, many questions remain related to when and how digital health technologies can be used to enhance measurement sensitivity and generate clinical insight. However, publicly available algorithms for processing data are limited, especially for task-based assessment of bradykinesia. To this end, we describe our algorithm methodology and include pseudocode so that other researchers may adapt, improve, and test our methodology on other datasets. We have added clarifying text to the Discussion section (first and third paragraphs) of the manuscript reflecting our viewpoint.

3. To what degree the sensitivity was improved by the proposed score?

Thank you for your important question. We chose to evaluate the differences in sensitivity using paired two-sided Wilcoxon rank-sum tests. We believe this analysis methodology is useful for determining whether there is a statistical difference between the two time points evaluated in the study, and thus a change in symptom impairment between baseline and 12 months. Although it is difficult to make a direct comparison of the degree of sensitivity between the MDS-UPDRS, a short ordinal scale ranging between 0 and 4, and the continuous digital composite scores, ranging from approximately -6 to 21, we believe that the use of standardized effect size and the proportion of participants whose score increased from baseline to month 12 help to clarify the degree of change.

Therefore, in order to provide further information related to the degree of change between the two visits, we have added statistical details to the manuscript, including Wilcoxon test statistics, confidence intervals, Cohen's d standardized effect sizes, and the proportion of participants whose score increased from baseline to month 12. We have adjusted the Results and Methods sections to include these analysis methods.

4. The authors displayed the comparison of “baseline VS 12 months” were compared, how about baseline VS 3 months, 6 months, and 9 months?

Thank you for your question. Baseline versus 12 months was the focus of the manuscript as 12 months was the minimum amount of time to measure strong changes in bradykinesia severity using digital measures in our study population. However, we have added the results of a sub-analysis comparing baseline versus 9, 6, and 3 months within the Results section (4th paragraph of the 1st section). The digital composites trend toward distinguishing between baseline and 9 months, though 12 months is required to measure statistically significant change.

5. The digital score and the clinical measurement may describe the same thing, they may not independent. In the reviewer's view, the digital score is the objectification and quantification of the MDS-UYPDRS score. Therefore, can Kruskal-Wallis test be applied for comparison?

Thank you for your question. Although the digital score and clinical measures are both describing the phenomenon of bradykinesia impairment, they are independent measures. For example, when assigning an MDS-UPDRS score, clinicians do not have access to digital measures, and their ratings are solely based on their expert judgement. Likewise, clinician ratings are not used to derive or influence the values of digital measures. Thus, both MDS-UPDRS scores and digital measures presented here are independent as the occurrence of one does not affect the probability of occurrence of the other. Furthermore, there is some precedence in the literature for using Kruskal-Wallis test as a method to compare between digital measures and clinical scores (Mahadevan et al. 2020; DOI: 10.1038/s41746-019-0217-7). In addition to Kruskal-Wallis, we include Spearman correlation to evaluate the relationship between the two variables as well as unpaired two-sided Wilcoxon rank-sum tests for comparisons between discrete clinical scores. If this explanation is not sufficient, we are glad to consider making changes based on feedback.

6. How the test-retest reliability was defined? It looks like in the longitudinal comparison? Since the disease is developing all the time, it's normal that the retest in 1 month varies from the test in baseline. Test-retest may be used to describe the tests of same situation, so it may be confusing in this manuscript.

Thank you for your questions. Ideally, from a statistical standpoint, test-retest reliability would be conducted on as short a timescale as possible to limit any changes due to disease progression or other extraneous factors that may affect performance. However, test-retest periods of two weeks are typically recommended as the 'standard' cadence of repeat assessments (Streiner et al. 2014; DOI: 10.1111/jan.12402), and in fact the primary study examining the multisite test-retest

reliability of the MDS-UPDRS in early PD patients had a mean test-retest duration of 14.6 days (range 3-36 days; Siderowf et al. 2002; DOI: 10.1002/mds.10011) which is not far from the duration between measures in the current study. In addition, for this study, in which participants must spend several hours at the clinic to perform various procedures related to the MDS-UPDRS, it was not feasible to repeat assessments twice at the same visit. Despite the 1 month difference between visits, in the case of early Parkinson's disease, large changes in disease severity are unlikely to occur over such short timescales, and indeed our own results suggest 12 months is required to see statistical changes in bradykinesia severity. Thus, the 1-month timepoint was used to evaluate test-retest reliability as minimal changes due to disease state would have occurred and we feel the value of including this analysis in the manuscript supersedes the limitations, as major fluctuations at one month would indicate unreliable, spurious measurements, calling into question the validity of the digital measures. We have added text to the Results section (1st paragraph of the section titled 'Digital composites demonstrate moderate test-retest reliability') and the Discussion section (5th paragraph) to clarify our reasoning for the test-retest reliability analysis.